# Prevalence of Dietary Supplement Use among Athletes Worldwide: A Scoping Review

**DOI:** 10.3390/nu14194109

**Published:** 2022-10-03

**Authors:** Jana Daher, Moriah Mallick, Dalia El Khoury

**Affiliations:** Department of Family Relations and Applied Nutrition, University of Guelph, 50 Stone Road, Guelph, ON N1G 2W1, Canada

**Keywords:** dietary supplements, athletes, prevalence, sports nutrition, reasons for use, sources of information

## Abstract

Athletes represent a major part of dietary supplement users. This scoping review aims to explore the prevalence of dietary supplement use among athletes worldwide, most commonly used supplements, sources of information on dietary supplements and their reasons for use of these supplements. PubMed, CINAHL, MEDLINE, and PsycInfo were searched for original research articles. Studies were included if they involved athletes, identified the prevalence of dietary supplement use, and were published after 2017. A total of 26 articles were reviewed. Prevalence of dietary supplement use varied among articles, but sex-based differences related to the types of used dietary supplements existed. Generally, the findings were consistent in terms of reasons for use and sources of information. Unfortunately, the lack of homogeneity regarding the definition of dietary supplements, definition of use, reporting timeframes, and data collection methods complicates the attempt to compare the findings among studies.

## 1. Introduction

Natural health products, often referred to as dietary supplements, are naturally occurring substances intended to supplement the diet to restore or maintain good health [1]. These commercially available substances include, but are not limited to, proteins, herbs, botanicals, vitamins, and minerals [2]. The absence of consensus on a clear definition or categorization of dietary supplements [3,4,5,6] can complicate the attempts to provide an accurate overview of the current state of prevalence, posing multiple challenges to the interpretation of relevant research [7]. For example, the International Olympic Committee Consensus Statement defined dietary supplements as, “A food, food component, nutrient, or non-food compound that is purposefully ingested in addition to the habitually consumed diet with the aim of achieving a specific health and/or performance benefit”, [8], and the National Institutes of Health defines dietary supplements as, “A product that is intended to supplement the diet; A dietary supplement contains one or more dietary ingredients (including vitamins, minerals, herbs or other botanicals, amino acids, and other substances) or their components; is intended to be taken by mouth as a pill, capsule, tablet, or liquid; and is identified on the front label of the product as being a dietary supplement” [9].

Substantial research has been conducted to study the prevalence of dietary supplement use and relevant practices around the world. Athletes were one of the most targeted groups of interest because they represent a major part of dietary supplement users [10]. Previous work has estimated the prevalence of dietary supplement use among athletes to be between 40% and 100%, depending on several factors including the level of competition, type of sport, and the definition of dietary supplement use [7].

This raises the concern of dietary supplement misuse and inadvertent doping (i.e., when a prohibited substance is unknowingly consumed) as they can expose users to harmful substances or precursors of prohibited substances. Initiated by the International Olympic Committee in 1999, the World Anti-Doping Agency (WADA) was established, commencing the fight against doping in sports. Due to being classified as a subcategory of food, manufacturers of dietary supplements are not required to provide consumers with evidence of product safety or efficacy before marketing their products as they are exempt from Federal Drug Administration approval [7]. This means that manufacturing companies do not have to test their products for banned substances according to the WADA list. This can pose great risks to athletes’ careers. For example, in September 2019, Carina Horn, a South African Olympic athlete, failed a drug test due to contaminated supplements and was sentenced to a 4-year ban. Luckily, Horn fought the claims against her in court and won her case. A scientist confirmed that the prohibited substances came from supplements Horn was consuming, more specifically, contaminated pre-and post-workout supplements. This incident of inadvertent doping caused Horn to miss the World Championships in 2019, and Tokyo’s Summer Olympics in 2021 [11]. Similarly, in April 2019, Brandon Copeland, an American football linebacker, also failed a drug test due to contaminated supplements. Though unintentional, this incident resulted in a four-game suspension and a $745 fee to test each supplement to expose the contaminated product [12].

Contamination can occur either due to inadequate manufacturing procedures or can be intentional by manufacturers to increase the effectiveness of their supplements [13]. The most frequently reported undeclared contaminants of dietary supplements are anabolic-androgenic steroids and stimulants [14], which are mostly found in supplements targeted for enhancing athletic performance [13]. This amplifies the importance of purchasing third-party tested supplements. These are supplements tested for their safety and purity by third-party programs [15]. Unfortunately, many dietary supplements do not undergo third-party testing, and therefore, dietary supplements remain a great threat to competing athletes with the risk of contamination with doping-related substances [15].

Given that common sources of dietary supplement-related information come from fellow teammates, coaches, the internet, family and friends or their judgment [6], an educational intervention program is a possible way to improve dietary supplement knowledge among athletes to decrease the risk of dietary supplement misuse and inadvertent doping [16]. Considering the many risks associated with dietary supplement consumption, a food-first approach is often recommended to all populations when considering the use of dietary supplements. However, it is important to understand that this approach may not always be practical or appropriate, especially for athletes [17]. There are many reasons why the risk-benefit analysis may favor the consumption of dietary supplements. These include: (1) certain nutrients are difficult to consume through the diet in sufficient amounts, (2) some nutrients are limited to food items an athlete might not eat, (3) uncertainty of the number of ergogenic nutrients consumed due to high variability in food items, (4) larger doses of concentrated nutrients needed to correct deficiencies or other health implications, (5) whole foods may be impractical to consume before, during, or after physical activity and (6) dietary supplements may help when contamination of food hygiene is a concern [17]. Therefore, Close et al. (2022), suggest ‘food first’ should mean, “where practically possible, nutrient provision should come from whole foods and drinks rather than from isolated food components or dietary supplements”. The International Association of Athletics Federations Consensus Statement [18] and the International Olympic Committee Consensus Statement [8] also acknowledge that a food-first approach may not always be practical, as the use of dietary supplements may be necessary for athletes to meet their nutritional needs when whole food consumption is impractical due to training schedules, preparation and storage issues and gut comfort [8,18]. Considering the emerging concerns regarding dietary supplements’ safety, several studies have investigated their integrity and authenticity. Furthermore, the attempt to compare the prevalence of dietary supplement use and practices worldwide might not be so accurate due to inconsistent definitions among studies and a lack of differentiation between the general categories of dietary supplements, which complicates the attempts to compare specific trends, especially among athletes. This does not provide specificity to be able to compare studies on specific dietary supplements used by athletes. Currently, there are no recent rigorous research reviews that explore the prevalence of supplement use among athletes. Therefore, the purpose of this scoping review was to explore the prevalence of dietary supplement use among athletes worldwide from 2018 to 2022, most commonly used supplements, reasons for dietary supplement use and sources of information.

## 2. Materials and Methods

This scoping review was designed and conducted in accordance with the Preferred Reporting Items for Systematic Reviews and Meta-Analyses Extension for Scoping Reviews (PRISMA-ScR) guidelines (Appendix A) [19]. The review protocol can be obtained upon request from the corresponding author.

### 2.1. Eligibility Criteria

To be included in the review, peer-reviewed journal papers needed to (1) involve athletes, (2) identify the prevalence of dietary supplement use among athletes, (3) be published after 2017. Studies were excluded if they (1) investigated supplement use in gym users, (2) collected data before 2016. All the studies had to be fully completed and published; abstract-only, presentation-only, and unpublished studies were excluded.

### 2.2. Information Sources

Searches were conducted on the electronic databases of PubMed, CINAHL, PsycInfo, and MEDLINE for studies published from 2018 up until June 2022 (The last search was conducted in July 2022). The search was restricted to papers written or translated to English. Reference lists of all retrieved articles in addition to the profiles of authors with extensive experience in dietary supplement research were scanned for additional relevant articles.

### 2.3. Search Strategy

The search strategy involved a combination of keywords including (supplement OR product OR vitamin OR mineral OR protein OR ergogenic OR drink OR herbal OR amino acid OR nutraceutical OR bar), AND (athlete OR player OR physically active OR elite OR sport OR bodybuilder OR runner OR team OR football OR soccer OR swimmer OR weightlifter OR dancer OR triathlete OR baseball OR gymnastic OR ballet OR tennis OR sailor OR basketball OR hockey), AND (use OR consumption OR prevalence OR survey OR pattern OR report OR intake OR habit). The search was limited to results in which these keywords show in the title or abstract.

### 2.4. Selection Process

Studies were screened for eligibility by two of the authors. Any disagreements on study selection or data extraction were resolved by consensus or after discussion with the third author.

### 2.5. Data Charting and Items

The data charting table was developed by the first author and further improved and approved by the other two authors to finally include the authors’ names, year of publication, country where each study was conducted, supplements of focus, type of sport, target population, age, sample size, data collection method, and results (prevalence of supplement use, most commonly used supplements, reasons for use, and sources of information).

## 3. Results

### 3.1. Selection of Studies

A total of 596 studies were identified through searching databases and other sources. After removing duplicates (n = 105), 491 articles were screened based on their titles and abstracts only, yielding 52 studies for full-text screening. Twenty-six articles were excluded, leaving a final pool of 26 studies that were eligible for inclusion in this scoping review (Figure 1).

### 3.2. Characteristics of Studies

Figure 2 represents the studies’ demographics. Studies were mostly conducted in Spain (31%) and the United States of America (27%) [15,20,21,22,23,24,25,26,27,28,29,30,31,32,33]. Some studies investigated supplement use in more than one country [34,35,36,37]. The twenty-six studies included a total of 17,342 participants. The sample sizes ranged from 20 [26] to 2113 subjects [24]. The majority of the studies included adult athletes, some included under 18 athletes [20,24,25,28,32,34,38,39,40], and none included older-adult athletes. Most studies included athletes from several sport categories [6,15,20,21,22,27,31,34,36,37,38,40], while others focused on one type of sport such as running [23,24], bodybuilding [41,42], football [32,35], rowing [26], sailing [25], rugby [30], fencing [28], handball [29], squash [33], and track and field [39]. Interestingly, one study focused on paralympic sports [34].

### 3.3. Data Collection Methods and Reporting Timeframes

All studies used self-reported questionnaires or surveys to collect data on dietary supplement use among athletes. Some studies have provided their questionnaire or survey in an online format [24,27,29,32,37,40,41]. However, the selected reporting timeframe varied across studies (Figure 3). Most of the studies investigated current use of dietary supplements, while other timeframes were the past year (n = 7), past six months (n = 2), past 3 months (n = 1), past month (n = 1), and past season (n = 1). None of the studies used qualitative or mixed-methods approach to research.

### 3.4. Prevalence of Dietary Supplement Use among Athletes

The characteristics of each study, prevalence of dietary supplement use and most used dietary supplements are summarized in Table 1. It is worth highlighting that the defined use of dietary supplements differed among studies. A large number of the studies defined use as reporting using at least one dietary supplement in the chosen timeframe [6,15,20,21,23,29,30,35,40,43]. Other studies defined use as using at least one dietary supplement on two or more days per week [22,24,31]. The rest of the studies did not clarify how they defined dietary supplement use.

The prevalence of dietary supplement use is presented in Table 1. The included studies covered a wide range of individual and team sports. Most studies included athletes from several sport categories, while others focused on one type of sport such as running, rowing, sailing, bodybuilding, rugby, fencing, handball, football, squash, and track & field.

Prevalence ranged from 11% [37] to 100% [15,26,33]. The use of dietary supplements among athletes was relatively high in the majority of the studies, but this can vary based on the breadth of the definition of a dietary supplement, reporting timeframe (i.e., past month vs. past year), data collection method and format, and definition of dietary supplement use (Table 1). In general, the consumption of dietary supplements was higher in males compared to females except for few studies [23,24,39]. Protein and vitamin/mineral supplements were among the most commonly used supplements reported by athletes. In most of the studies that presented the most commonly used supplements based on sex, it was clear that females were more likely to use vitamin/mineral supplements compared to males, while males were more likely to use protein supplements.

### 3.5. Reasons for Dietary Supplement Use

The reported reasons of dietary supplement use were generally consistent across studies (Table 2). The most frequently reported reasons are improving athletic performance, improving health, and accelerating recovery. Very few studies, among those that investigated motivations behind dietary supplement use, presented sex-based differences [22,25,34]. Females were more likely to use dietary supplements for health reasons, while males were more likely to report using dietary supplements to enhance performance.

### 3.6. Sources of Information on Dietary Supplements

Sources of information on dietary supplements were investigated in most of the studies (Table 3). Health care professionals, coaches/trainers, the internet, and teammates were among the most commonly reported sources of information. It is apparent that males are more likely to rely on their coach/trainer, teammates, dietitian/nutritionist, and family and friends to receive information on dietary supplements, while females are more likely to rely on doctors/healthcare professionals, their coach/trainer, and family & friends.

## 4. Discussion

The aim of this scoping review was to explore the prevalence of dietary supplement use among athletes worldwide in the past 5 years, most commonly used supplements, reasons for dietary supplement use and sources of information. Using rigorous scoping review methods, 27 studies were reviewed to examine the current breadth of knowledge of dietary supplement prevalence among athletes; no qualitative or mixed-methods studies were found. Given the increasing consumer interest in health and well-being that is well-reflected by the fast-growing dietary supplements industry, which is expected to reach around USD 278 billion by 2024, it is vital to examine the prevalence of dietary supplement use among athletes [44]. In this current review, the prevalence ranged between 11% [37] to 100% [15,26,33]. In general, the prevalence of dietary supplement consumption was higher in males compared to females except for few studies [23,24,39]. Additionally, there was minimal reporting on age-based differences, however few studies noted age as an influence on dietary supplement consumption, with older athletes consuming more dietary supplements than younger ones [20,27,39].

Currently, there is no universal definition of dietary supplements used within the scientific community, therefore, the definition used to define dietary supplements varied across studies. The broadness of definitions used to describe dietary supplements by studies affect their findings’ comparability with other studies that define dietary supplements differently. Many authors studying this subject area also noted difficulty in comparing studies due to the lack of homogeneity between definitions and categorizations of dietary supplements [2,7]. The absence of a standard definition for dietary supplements creates an issue within the scientific community to accurately study trends between the results of different studies.

Furthermore, it is important to note that the reported prevalence by each study included in this review heavily relied on how they defined dietary supplement use. While some studies defined it as using at least one dietary supplement in a chosen timeframe [6,15,20,21,23,29,30,35,40,43], others defined it as using at least one dietary supplement on two or more days per week [22,24,31]. The remaining studies did not explicitly define dietary supplement use. Along with inconsistencies between defined dietary supplement use among studies, inconsistencies between used time frames also complicates the attempt to compare the prevalence of dietary supplement use among athletes. Current use of dietary supplements, use of dietary supplements in the past year, past six months, three months, past month, and past season were all timeframes used by studies. Similarly, lack of consistent methodology also hinders researcher’s ability to compare studies. Interestingly, some studies have used online questionnaires/surveys to explore dietary supplement use, and all of these were conducted after 2019, which implies how the COVID-19 pandemic has affected this type of research, and might shape the format of similar future studies.

Most studies in this review focused on a wide range of dietary supplements, while only one focused on a specific dietary supplement such as tart cherry [37]. A possible reason dietary supplements may seem attractive to athletes is that they are continuously seeking ways to gain advantages over their competitors. Based on the studies included in this review, it appears the main reasons for dietary supplement use are improving athletic performance, improving health, and accelerating recovery. Fascinatingly, one study included in this review surveyed the attitudes of athletes who refrain from dietary supplement use, with the main reasoning being that they do not need them, and that they do not know enough about them [34]. In the few studies that examined sex-based differences, it was concluded that females were more likely to consume dietary supplements for health reasons and males were more likely to consume dietary supplements to enhance performance [22,25,34]. Another sex-based difference one study found is that females often used dietary supplements ‘at times’ rather than on a regular basis [43]. Potential false and misleading advertisement by companies in the dietary supplement industry coupled with lack of knowledge put athletes at great risk of inadvertent doping [32]. It is alarming that there is a large number of studies in which athletes reported that they rely on their coaches, teammates, the internet or family and friends for information on dietary supplements. Referring to a reliable source of information with regard to dietary supplements is critical in the athletic community, considering the probability of supplement contamination with prohibited substances and consequent health risks and anti-doping sanctions.

Determining the prevalence of doping amongst athletes was not within the scope of this review. However, considering the great risk of dietary supplement contamination with prohibited substances, athletes should consider all other options before obtaining nutrients from dietary supplements [7]. Consuming nutrients from whole foods provides greater benefits than the consumption of isolated nutrients [45]. Dietary supplements cannot excuse inadequate diets, and therefore, it is recommended that athletes optimize and prioritize their nutrient intake from whole foods. Of course, dietary supplement use cannot be fully disputed as there are circumstances where dietary supplement consumption is necessary [17]. Dietary supplements may be of great value to athletes who are vegan, vegetarian, or are treating deficiencies [17].

Despite the high prevalence of athletes consuming dietary supplements, there was minimal reporting on whether the dietary supplements were third party tested. Only one study reported information on this and found that while over 90% of participants believed it was vital to know whether the dietary supplement was third party tested, only 57% purchased third-party tested supplements [15]. Knowledge about nutrition and dietary supplement use is vital to athletes choosing to consume dietary supplements. Accurate information is needed to make informed decisions and protect against inadvertent doping and dietary supplement misuse. The integration of nutritional-knowledge programs may be highly valuable to increase the knowledge, beliefs and intentions of athletes who choose to consume dietary supplements [16].

A wide range of team and individual sports were included in the studies within this review. Interestingly, one study focused on wheelchair sport teams [34]. Aside from reasons for dietary supplement use, another sex-based difference observed was generally, dietary supplement consumption was higher in males than females. Only a few studies found contradictory results [23,24,39]. Gender minorities, including transgender individuals and gender non-binary individuals, were not included in this analysis, as no articles targeted this population. Future studies should include and study minority populations and paralympic sports to increase the generalizability of their results.

There are a number of limitations for the studies conducted to date on this topic. Firstly, a combined 58% of studies included within this review were conducted in Spain and the United States of America: mainly, by the same research team. Little research was conducted in the United Kingdom, East Asia and Middle Eastern countries. Moreover, there were no qualitative reviews on this topic. Qualitative studies provide a new perspective and can provide great insight that allows for better understanding and interpretation of results. Studies included in this review relied on self-assessment measures for dietary supplement intake information. Using self-reported data is subject to bias and is limited to the participants ability to assess themselves accurately.

This scoping review does have some limitations. First, our search was limited to articles written in or translated to English language, so there is a possibility that relevant articles written in other languages were left out. Additionally, the included studies were not assessed for risk of bias, which is critical to identify the quality of evidence. Nevertheless, this review provides a comprehensive overview of the recent prevalence of dietary supplement use among athletes.

## 5. Conclusions

In conclusion, the lack of homogeneity among studies regarding the definition of dietary supplements, reporting timeframes, and data collection methods complicates the attempt to compare the findings. Results from this review may contribute to future educational intervention program initiatives to increase the knowledge of users on dietary supplements, their benefits, and their risks and on the importance of their use under professional guidance. Given the higher prevalence of use of dietary supplements among athletes, more research and initiatives are indispensable in this population as proper nutrition is an extremely important factor in ensuring a high quality of life and optimum performance for athletes.

## Figures and Tables

**Figure 1 nutrients-14-04109-f001:**
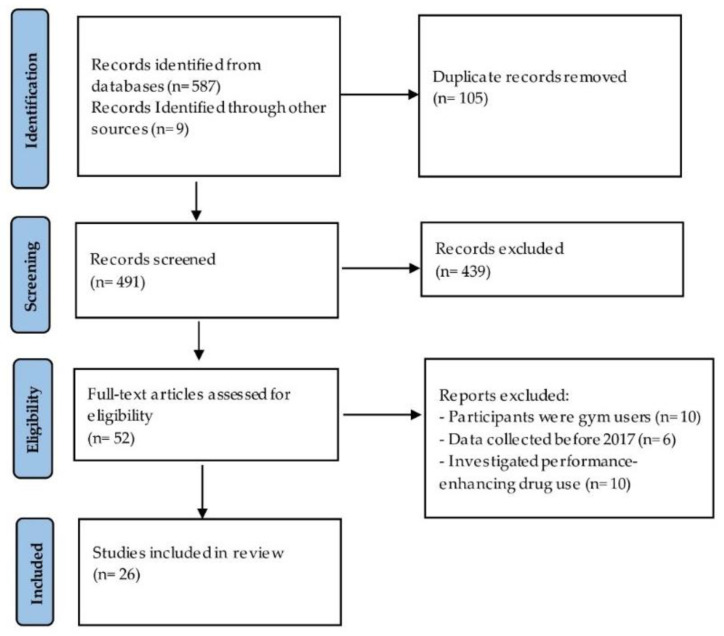
Selection of sources of evidence.

**Figure 2 nutrients-14-04109-f002:**
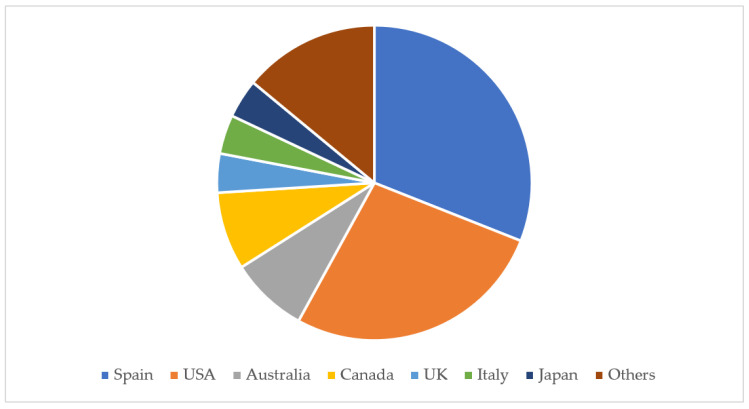
Summary of studies’ demographics.

**Figure 3 nutrients-14-04109-f003:**
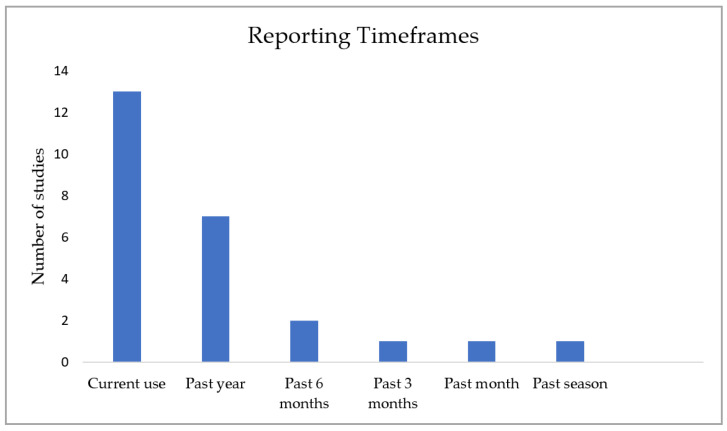
Reporting timeframes across studies.

**Table 1 nutrients-14-04109-t001:** Summary of studies with prevalence and most used dietary supplements.

Publication-Country	Target Population	Age Mean ± SD (Years)	Sex (Sample Size)	Data Collection Method (Time Frame)	Prevalence (Defined Use)	Most Used Supplements
Aguilar-Navarro et al. (2021) Spain [20]	Elite athletes (Individual & team sports)	15–66	Total: 504 M (*n* = 329) F (*n* = 175)	Questionnaire (preceding season)	62% M:65% F: 57% (reported using at least one DS)	M: Protein supplements. F: Multivitamins; Branched chain amino acids.
Baltazar-Martins et al. (2019) Spain [21]	Elite athletes (Individual & team sports)	NC	Total: 527 M (*n* = 346) F (*n* = 181)	Questionnaire (past year)	64% M: 67% F: 58% (reported using at least one DS)	Proteins; amino acids/ Branched chain amino acids; multivitamins.
Barrack et al. (2022) USA [22]	NCAA Division I athletes	NR	Total: 557 M (*n* = 298) F (*n* = 259)	Survey (past year)	45% (reported using at least one DS on 2 or more days per week)	M: Protein/amino acid supplements. F: Vitamin/mineral supplements.
Barrack et al. (2021) USA [23]	Elite collegiate endurance runners	18–22	Total: 135 M (*n* = 65) F (*n* = 70)	Survey (past 4 weeks)	79% M: 74% F: 83% (reported using at least one DS)	Multivitamin/minerals; iron. M: Amino acids; beta-alanine. F: Iron; calcium.
Barrack et al. (2020) USA [24]	Preadolescent endurance runners	13.2 ± 0.9	Total: 2113 M (*n* = 1255) F (*n* = 858)	Web-based Survey (past year)	26% M: 22% F: 33% (reported using DS on 2 or more days per week)	Sport foods; multivitamin/minerals. M: Creatine and sport foods. F: Multivitamin/minerals, vitamin D, calcium, iron, probiotic supplements, and diet pills.
Caraballo et al. (2020) Spain [25]	Elite sailors	12–17	Total: 42 M (*n* = 31) F (*n* = 11)	Questionnaire (General and current)	50% M: 55% F: 46%	M: Isotonic drinks; caffeine. F: Vitamin D; vitamin complexes.
Domínguez et al. (2020) Spain [26]	Heavyweight and lightweight rowers	23 ± 3	Total: 20 M (*n* = 16) F (*n* = 4)	Questionnaire (general and current—during the sports season)	100%	Iron; caffeine; β-alanine, energy bars; vitamin supplements; and isotonic drinks.
Graybeal et al. (2022) USA [27]	Endurance cyclists, runners, and triathletes.	39.4 ± 13.5	Total: 200 M (*n* = 92) F (*n* = 108)	Digital questionnaire (current use)	78%	Multivitamin; electrolytes; vitamin D; protein.
Hackett (2022) Australia [41]	Bodybuilders	≥18 years	Total: 235 M (*n* = 235)	Online survey (off season and 6 weeks before a competition)	96%	Creatine monohydrate; whey protein.
Hurst et al. (2020) United Kingdom [38]	Team and individual sports athletes	20.8 ± 4.5	Total: 557 M (*n* = 429) F (*n* = 128)	Survey (current use)	53%	Ergogenic supplements.
Jovanov et al. (2019) Serbia, Germany, Japan, Croatia [34]	Team and individual sports athletes	15–18	Total: 348 M (*n* = 174) F (*n* = 174)	Survey (current use)	82% M: 61% F: 39%	M: Whey protein, creatine, amino acids, caffeine, and NO reactor. F: Vitamins and mineral complexes.
Madden et al. (2018) Canada [43]	Wheelchair rugby athletes	36.3 ± 9.5	Total: 42 M (*n* = 33) F (*n* = 9)	Questionnaire (past three months)	M: 91% F: 78% (reported using at least one DS)	Electrolytes, sport bars, vitamin D, protein powder, and MVMM (multivitamin multimineral). M: Vitamin D, protein powder, and electrolytes. F: MVMM and vitamin D.
Mata et al. (2021) Spain [28]	Fencers	21.8 ± 5.9 years	Total: 49 M (*n* = 18) F (*n* = 31)	Questionnaire (General and current)	47%	Sports drinks, vitamin C, sport bars, caffeine. M: Sports drinks, sports bars, and iron. F: Sports drinks, sports bars, and caffeine.
Montuori et al. (2021) Italy [42]	Bodybuilders	>18	Total: 107 M (*n* = 73) F (*n* = 34)	Questionnaire/survey (general and current)	82% M: 66% F: 34%	NR
Muñoz et al. (2020) Spain [29]	Handball players	NR	Total: 187 M (*n* = 112) F (*n* = 75)	Online Questionnaire (current use)	60% (reported using at least one DS)	Sports drinks, energy bars and caffeine-containing products. M: creatine and L-carnitine.
Oliveira et al. (2022) Australia, Canada, Iceland, Netherlands, Norway and Portugal. [35]	Elite football players	Median age: 24	Total: 103 F (*n* = 103)	Questionnaire (past year)	82% (reported using at least one DS)	Vitamin D, omega-3 fatty acids, and protein (including whey protein and casein).
Roy et al. (2021) Canada [6]	Varsity athletes	20.5 ± 1.8	Total: 302 M (*n* = 92) F (*n* = 210)	Questionnaire (past 6 months)	58% M: 66% F: 53% (reported using at least one DS)	Protein, vitamins, minerals, and carbohydrate supplements. M: Amino acid supplements and stimulants. F: Prebiotics and probiotics.
Sánchez-Oliver et al. (2021) Spain [30]	Rugby players	M: 24.3 ± 5.0 F: 24.0 ± 4.9	Total: 144 M (*n* = 83) F (*n* = 61)	Questionnaire (general)	65% M: 77% F: 49% (reported using at least one DS on some occasion)	Whey protein, caffeine, sport drinks, energy bars, creatine monohydrate, BCAAs, and glutamine.M: Whey protein, creatine monohydrate, and glutamine.
Sassone et al. (2019) USA [31]	NCAA Division I athletes	18–26	Total: 557 M (*n* = 229) F (*n* = 258)	Survey (past year)	45% (reported using ≥1 dietary supplements ≥2 days per week)	Pre-workout & herbal supplements.
Sekulic et al. (2019) Croatia and Kosovo [36]	Professional team-sport athletes	22.11 ± 3.37	Total: 912 M (*n* = 556) F (*n* = 356)	Questionnaire (General and current)	13% (consumed DS regularly 36% (consumed DS occasionally)	Vitamins/minerals, isotonic drinks, energy bars, iron, recovery supplements, carbohydrates, proteins/amino acids.
Shoshan et al. (2021) USA [32]	Football players	16.9 ± 1.2	Total: 102 M (*n* = 98) F (*n* = 4)	Online questionnaire (general and current)	60% (protein supplements)29% (pre-workout supplements)	NR
Tabata et al. (2020) Japan [39]	Track and field elite athletes	Junior athletes: 17.7 ± 1.1 years Senior athletes: (25.2 ± 3.9 years	Total: 574 M (*n* = 314) F (*n* = 260)	Pre participation medical form (current use)	64% M: 60% F: 69%	Amino acids, vitamins, minerals, proteins. M: Protein, creatine. F: Vitamins, amino acids.
Vento & Wardenaar (2020) USA [15]	NCAA I collegiate student athletes	20 ± 1.6 years	Total: 138 M (*n* = 49) F (*n* = 89)	Questionnaire (Past year)	100% (reported using at least one DS)	Multivitamin and mineral supplements, and single vitamins or minerals. F: Vitamins and single minerals, exotic berries, herbs, maca root powder, ribose, ephedra, colostrum, and hydroxy-methyl-butyrate.
Ventura Comes et al. (2018) Spain [33]	National & international squash players	International players: 25.0 ± 6.2 National players: 35.6 ± 14.2	Total: 42 M (*n* = 29) F (*n* = 13)	Questionnaire—survey (General and current)	lnternational athletes: 100% National athletes: 68%	Ergogenic aids C, sports food.
Waller et al. (2019) Australia [40]	Individual and team sports’ athletes	20.4 ± 4.5	Total: 94 M (*n* = 39) F (*n* = 55)	Online questionnaire (past year)	87% (reported using at least one DS)	Sports drinks, caffeine, protein powder, and sports bars.
Wangdi et al. (2021) ^a^ 15 countries [37]	Individual and team sports’ athletes	27.6 ± 9.8	Total: 80 M (*n* = 51) F (*n* = 27) did not disclose (*n* = 2)	Online questionnaire (current and previous use)	11% (reported current use)11% (reported previous use)	N/A

Percentages were rounded to the nearest whole number. DS: Dietary supplements; F: Females; M: Males; NR: Not reported; NC: Not clear; ^a^ This study investigated tart cherry use only.

**Table 2 nutrients-14-04109-t002:** Reasons for dietary supplement use reported by athletes.

Publication	Reasons for Dietary Supplement Use (% of Participants that Reported the Reason *)
Barrack et al. (2020) [24]	M: Increasing strength/power, increasing muscle mass. F: Improving health
Carabello et al. (2020) [25]	M: Improving performance (65%) and physical appearance (15%). F: Improving health status (57%), preventing nutritional deficits (14%).
Domínguez et al. (2020) [26]	Improving recovery (80%), health reasons.
Graybeal et al. (2022) [27]	Improving performance and health, meeting nutrient requirements.
Jovanov et al. (2019) [34]	M: Improving athletic performance (19%) F: Improving health (18%)
Madden et al. (2018) [43]	Performance—medical/health
Mata et al. (2021) [28]	Improving performance (34%), improving health (29%)
Muñoz et al. (2020) [29]	Enhancing sports performance (54%), improving health (13%), and improving physical appearance (11%).
Oliveira et al. (2022) [35]	Staying healthy (66%), accelerating recovery (58%), increasing energy reducing fatigue (54%).
Roy et al. (2021) [6]	Maintaining good health (83%), increasing energy (71%), promoting recovery (69%), correcting or preventing micronutrient deficiencies (60%) and supplying convenient forms of energy and/or macronutrients (58%)
Sánchez-Oliver et al. (2021) [30]	Improving sport performance (62%), preventing nutritional deficits (14%)
Vento & Wardenaar (2020) [15]	Improving health and performance.
Waller et al. (2019) [40]	Enhancing recovery (63%), maintaining health (59%), and improving energy (50%)
Wangdi et al. (2021) [37]	Improving recovery (75%), sleep and immunity (30%), and general health (30%)

M: Males; F: Females; * Percentages were rounded to the nearest whole number; * Percentages were not reported for studies that split data into groups.

**Table 3 nutrients-14-04109-t003:** Sources of information on dietary supplements reported by athletes.

	Studies (% of Participants that Reported the Reason *)		
Source of Information	Reported by Males	Reported by Females	Reported by All Participants (In Case Sex-Based Differences Were Not Reported)
Doctor/health professional	Mata et al. (2021) [28] (50%);	Aguilar-Navarro et al. (2021) [20] (60%); Barrack et al. (2020) [24]; Sánchez-Oliver et al. (2021) [30] (16%);	Domínguez et al. (2020) [26] (50%); Graybeal et al. (2022) [27]; Montuori et al. (2021) [42] (64%); Muñoz et al. (2020) [29]; Oliveira et al. (2022) [35] (46%); Roy et al. (2021) [6] (59%); Vento & Wardenaar (2020) [15] (45%); Waller et al. (2019) [40];
Nutritionist/dietitian	Aguilar-Navarro et al. (2021) [20] (83%); Madden et al. (2018) [43] (52%); Sánchez-Oliver et al. (2021) [30] (19%);	Carabello et al. (2020) [25] (14%); Mata et al. (2021) [28] (22%);	Montuori et al. (2021) [42] (64%); Oliveira et al. (2022) [35] (43%); Vento & Wardenaar (2020) [15] (92%); Ventura Comes et al. (2018) [33] (21%);
Coach/trainer	Barrack et al. (2020) [24]; Carabello et al. (2020) [25] (42.3%); Madden et al. (2018) [43] (30%); Sánchez-Oliver et al. (2021) [30] (27%);	Aguilar-Navarro et al. (2021) [20] (85%); Mata et al. (2021) [28] (27%); Sánchez-Oliver et al. (2021) [30] (16%);	Domínguez et al. (2020) [26] (40%); Jovanov et al. (2019) [34] (41%); Muñoz et al. (2020) [29]; Oliveira et al. (2022) [35] (41%); Roy et al. (2021) [6] (39%); Vento & Wardenaar (2020) [15]; Ventura Comes et al. (2018) [33] (29%);
Internet/social media	Barrack et al. (2020) [24]; Madden et al. (2018) [43] (33%);	Madden et al. (2018) [43] (33%);	Graybeal et al. (2022) [27]; Jovanov et al. (2019) [34] (39%); Montuori et al. (2021) [42] (71%); Roy et al. (2021) [6] (48%); Waller et al. (2019) [40]; Wangdi et al. (2021) [37] (15%);
Teammates	Barrack et al. (2020) [24]; Carabello et al. (2020) [25] (23%); Mata et al. (2021) [28] (8%); Sánchez-Oliver et al. (2021) [30] (14%);	Madden et al. (2018) [43] (44%);	Roy et al. (2021) [6] (45%)
Family/friends	Carabello et al. (2020) [25] (19%); Mata et al. (2021) [28]; Sánchez-Oliver et al. (2021) [30] (17%);	Barrack et al. (2020) [24]; Carabello et al. (2020) [25] (43%); Mata et al. (2021) [28] (39%);	Roy et al. (2021) [6] (53%);
Self-education	Aguilar-Navarro et al. (2021) [20] (48%);	Aguilar-Navarro et al. (2021) [20] (28%);	Baltazar-Martins et al. (2019) [21]; Roy et al. (2021) [6] (48%); Sekulic et al. (2019) [36]
Scientific research			Graybeal et al. (2022) [27]; Wangdi et al. (2021) [37] (33%);

* Percentages were rounded to the nearest whole number. * Percentages were not reported for studies that split data into groups.

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
