# Peer review of "Prevalence of Dietary Supplement Use among Athletes Worldwide: A Scoping Review"

_nutrients, 2022, doi:10.3390/nu14194109_

Round 1

Reviewer 1 Report

 Dear Authors,

 Thanks for giving me the chance to read this manuscript, “Prevalence of Dietary Supplement Use Among Athletes Worldwide: A Scoping Review”. The current paper tries to explore the prevalence of dietary supplement use among athletes worldwide, their sources of information on dietary supplements and their reasons for use of these supplements.

This is an interesting topic in the field of the sport nutrients. However, there are minor issues in the current manuscript that should be carefully addressed to be further considered.

 1.       Method

 ·       The study did not describe the process of systematic review, which should contain the number of original papers, the number of paper excluded in the process, etc. Authors are advised to clearly present how they performed the systematic review and the figure associated.

 2.       Language

 ·       There are many format flaws in the current version of the manuscript. For example, where is the caption of figure 1? The format of caption (Figure 2)? Authors are advised to revise the format carefully.

 To sum up, I personally like this paper. However, the problems should be addressed in order to be further considered. Hope these suggestions help.

Author Response

Comments from Reviewer 1:

  • Comment 1: The study did not describe the process of systematic review, which should contain the number of original papers, the number of paper excluded in the process, etc. Authors are advised to clearly present how they performed the systematic review and the figure associated.
  • Response: We thank the reviewer for their comment. A flow diagram has been added to the manuscript in addition to a paragraph that describes the selection of sources of evidence process in detail.

  • Comment 2: There are many format flaws in the current version of the manuscript. For example, where is the caption of figure 1? The format of caption (Figure 2)? Authors are advised to revise the format carefully.
  • Response: We thank the reviewer for their comment. The format has been revised. The figures have also been added at the end of the manuscript because captions are being hidden under the text in the document.

Reviewer 2 Report

This is a well written manuscript.  They address the research question, identifying the prevalence and reasons why athletes use supplements.  I also appreciate their recognition of the limitations of the current data/knowledge in this area.

Author Response

We thank the reviewer for their positive feedback on the manuscript.